# Pyrvinium Pamoate: Past, Present, and Future as an Anti-Cancer Drug

**DOI:** 10.3390/biomedicines10123249

**Published:** 2022-12-14

**Authors:** Christopher W. Schultz, Avinoam Nevler

**Affiliations:** 1Developmental Therapeutics Branch, Center for Cancer Research, National Cancer Institute, National Institutes of Health, Bethesda, MD 20892, USA; 2Jefferson Pancreas, Biliary, and Related Cancer Center, Department of Surgery, Thomas Jefferson University, Philadelphia, PA 19107, USA

**Keywords:** pyrvinium, cancer, mitochondria, drug repurposing, WNT pathway, cancer stem cells, oxidative phosphorylation

## Abstract

Pyrvinium, a lipophilic cation belonging to the cyanine dye family, has been used in the clinic as a safe and effective anthelminthic for over 70 years. Its structure, similar to some polyaminopyrimidines and mitochondrial-targeting peptoids, has been linked with mitochondrial localization and targeting. Over the past two decades, increasing evidence has emerged showing pyrvinium to be a strong anti-cancer molecule in various human cancers in vitro and in vivo. This efficacy against cancers has been attributed to diverse mechanisms of action, with the weight of evidence supporting the inhibition of mitochondrial function, the WNT pathway, and cancer stem cell renewal. Despite the overwhelming evidence demonstrating the efficacy of pyrvinium for the treatment of human cancers, pyrvinium has not yet been repurposed for the treatment of cancers. This review provides an in-depth analysis of the history of pyrvinium as a therapeutic, the rationale and data supporting its use as an anticancer agent, and the challenges associated with repurposing pyrvinium as an anti-cancer agent.

## 1. Introduction

Polyamines analogs and peptide/peptoid-like structures have been proposed as promising anti-cancer compounds [1,2,3,4,5,6]. Cyanine dyes share structural similarities to several lipophilic polyamines [7,8], polyaminopyrimidines [9], and mitochondrial-targeting peptoids [10]. Pyrvinium is a cyanine dye with multiple amino groups with similarities to some naturally occurring polyaminopyrimidine/polyamine compounds. Since its discovery nearly eight decades ago, pyrvinium has been used as a dye, a fluorescent probe, and as an anthelminthic. Despite widespread clinical use as an anthelminthic in the past, its use has declined since the 1980s with the advent of newer, more effective drugs. However, over the past two decades, there has been an increasing amount of evidence suggesting that pyrvinium has multiple anti-cancer effects. In this review, we aim to explore both the history of pyrvinium and the data supporting pyrvinium as a clinically relevant cancer therapy.

### Historical Use of Pyrvinium Pamoate

Pyrvinium is a fluorescent red cyanine dye, which was first described along with a series of other molecules in 1946 as part of U.S. patent number 2,515,912 held by Lare E.V. and Brooker L.G.S. It is a small lipophilic compound, with multiple amine groups and a positive charge at physiologic pH. In a series of articles between 1947 and 1959 describing the anthelminthic (targeting parasitic worms) activity of cyanine dyes, pyrvinium chloride salt and pyrvinium pamoate (also known as pyrvinium embonate, see Figure 1) were noted to be effective anthelmintics [11,12,13,14]. It was subsequently determined that pyrvinium pamoate (PP) had similar efficacy and less clinical toxicity than pyrvinium chloride [15]. The reduced toxicity of pyrvinium pamoate compared to pyrvinium chloride was likely due to reduced water solubility, which likely leads to decreased systemic gastrointestinal absorption [16]. PP thus became one of the primary treatments for various worm infestations for both human and veterinary use [14,17,18,19,20,21,22,23]. With the advent of newer anti-parasitic agents, the utilization of PP as an anthelmintic in the United States gradually declined in the 1970s–1980s [24,25,26]. While PP is still an FDA-approved agent in the United States, it is no longer clinically utilized or even easily available for purchase as a clinical grade drug. It has generally been replaced by other deworming agents based on compounds such as mebendazole and pyrantel pamoate. Despite this, PP is still accepted as a safe and effective deworming agent and is still available worldwide under commercial names including Pyrvin, Vanquin, Pirok, Pyrcon, and Molevac. 

Although the use of PP as an anthelminthic has dwindled over the years, since the early 2000s, interest in PP has surged due to its potential utilization to target other diseases-causing organisms at every level from viral [27,28] through bacterial [29,30,31], plasmodium [32], and fungi [33,34,35] to multicellular organisms [36] (Figure 2). PP has also shown significant potential in multiple disease types for the ability to reduce toxicity, promote wound repair, and inhibit fibrotic tissue development [37,38,39,40,41,42]. Perhaps most importantly, PP has received significant interest from cancer researchers due to its ability to target multiple types of human cancers, in particular cancer stem cells. 

The first published report of PP administration in cancer patients, though not as a cancer-therapeutic, was in 1971 when it was determined that children with malignant neoplasms could tolerate PP as a deworming agent [43]. In 1990, an adult patient with bronchial asthma was discovered to also have smoldering T-cell leukemia along with a severe worm infestation; the patient was treated with PP, which was well tolerated [44]. Regretfully, neither study assessed the effects, if any, of PP on the patients’ cancer burden. 

In 2004, PP was found to have exquisite cytotoxicity against cancer cells grown in low-glucose conditions [45]. Over the next two decades, multiple anti-cancer mechanisms of action (MOA) for PP were described; the two that were most frequently noted were the inhibition of the WNT pathway and the inhibition of mitochondrial function. Despite the clear interest in PP for cancer treatment, with over 100 papers published describing PP as an anticancer agent, it was not until 2021 that the first clinical trial with pyrvinium pamoate as an anticancer agent was initiated (NCT05055323 [46]). While ongoing, this trial is intended to assess the safety and absorption of PP, potentially allowing for this FDA-approved therapeutic to be repurposed for the treatment of numerous cancer types.

## 2. PP Mechanisms of Action as an Anti-Cancer Agent

Unsurprisingly for a small-molecule compound, PP has been shown to act through multiple mechanisms. The majority of the published articles assessing PP specifically as an anticancer therapeutic have focused on two main MOA, which appear to be at the root of PP action: inhibition of the WNT pathway and inhibition of mitochondrial function (Table 1).

### 2.1. WNT Signaling

The WNT/β-catenin signaling pathway is one of the essential pathways regulating the cell differentiation, development, and self-renewal of stem cells. As such, it has also been tightly associated with key events in cancer initiation and progression [89,90,91,92,93,94,95,96,97,98]. In 2010, Thorne et al. [48] found that PP inhibited the WNT signaling pathway at low nanomolar concentrations (in vitro and in vivo) in Xenopus laevis and in human colon cancer cells. They determined that PP bound and activated Casein kinase 1α (CK1α), thereby promoting the degradation of β-catenin, the primary effector of the WNT pathway. Supporting this work, Cui et al. [65], Li et al. [69], and Shen et al. [99] have all subsequently demonstrated that PP targeted the WNT pathway in a CK1α-dependent fashion in various cancer models. However, other studies have contested this mechanism and have suggested that PP acts through the upstream inhibition of the Akt/GSK-3β/β-catenin pathway [64,100]. 

In addition to pre-clinical studies assessing PP as a monotherapy in colon cancer [50,72], PP was found to act as a chemoprotective agent in the formation of intestinal polyps in an APC^min^ mouse model, presumably through its inhibition of the WNT pathway [47]. PP was also shown to synergize with the KRAS inhibitor salirasib in a WNT-dependent manner, describing a potentially actionable targeted combination of PP and salirasib in KRAS-driven colon cancers [49]. 

### 2.2. Mitochondrial Inhibition

Like many other cyanine dyes, and similar to mitochondrial-targeting peptoids [10], pyrvinium is a lipophilic cation, which allows it to preferentially target the mitochondria. Historically, the activity of PP as an anti-mitochondrial agent was linked to its efficacy at inhibiting protozoa including the plasmodium species associated with malaria [30,32,101]. To an extent, this was also linked to pyrvinium’s ability to inhibit various fungi [35,102]. From an evolutionary standpoint, the endosymbiotic theory holds that mitochondria were generated in a progenitor eukaryotic cell through endosymbiosis and that mitochondria maintain many similarities with bacteria [103]. Unsurprisingly, multiple antibiotics have displayed significant efficacy against cancer cells through inhibiting the mitochondria [104,105,106,107]. PP, interestingly, was first found to be an anti-mitochondrial agent, and only subsequently found to show significant activity against bacteria including Mycobacterium tuberculosis [29,30,31].

Multiple studies have implicated mitochondrial inhibition as a key factor in the activity of PP against cancer cells. Compelling support for this MOA lies in the facts that PP activity is increased in nutrient-poor microenvironments [45,78], and that mitochondrial DNA-depleted cell lines have shown significant resistance to PP therapy [73,75]. Several mechanisms of action have been suggested to explain the PP-dependent mitochondrial inhibition. A frequently proposed mechanism revolves around the inhibition of the mitochondrial respiratory complex I, with data showing PP doses of 0.1–1μM significantly inhibited complex I enzymatic activity, resulting in decreased ATP production and the increased production of reactive oxygen species (ROS) [75,79]. Tomitsuka et al. [77] have also shown that PP treatment of cancer cells (Panc-1, DLD-1, HepG2) resulted in the inhibition of the NADH–fumarate reductase system, a system primarily associated with cancer metabolism in dually hypoxic and hypoglycemic conditions, and increased activity of complex II (succinate–ubiquinone reductase). 

In a recent study by Schultz et al. [78], a new MOA for PP-mediated mitochondrial inhibition was proposed. In the study, PP was shown to result in the significant depletion of mitochondrially encoded RNA transcripts and the downregulation of electron transport chain proteins. Here it was found that PP, similar to other G-quadruplex binding agents, bound G-quadruplex structures, which are over-represented in mitochondrial DNA and profoundly inhibited global mitochondrial transcription [108,109].

Overall, it is likely that all of these actions can occur with PP treatment. Acute treatment of PP will lead to the inhibition of complex I and the electron transport chain in normoxic conditions and the inhibition of the NADH–fumarate reductase system in hypoxic conditions. However, prolonged treatment will cause more global dysregulation of mitochondrial function through the inhibition of mitochondrial transcription in all conditions.

### 2.3. Tumor Stemness

An important feature of tumor cell population self-renewal and survival after chemotherapy treatment is through a specialized sub-population of cancer-initiating cells or “cancer stem cells” [110,111,112]. Interestingly, a common MOA attributed to the anti-cancer effect of PP is the inhibition of cancer stem cells. Xu et al. [54] found that PP targeted tumor-initiating cells and impaired tumor self-renewal in breast cancer. In vivo PP treatment slowed tumor growth, decreased lung metastasis, and synergized with radiotherapy. Zhang et al. have shown the inhibitory effects of pyrvinium pamoate (PP) on lung cancer stem cells in vitro [55,56]. Similar studies have found PP to inhibit tumor stemness in multiple cancer types including melanoma, leukemia, glioblastoma, and cancers of the prostate, pancreas, lung, ovary, and breast [57,73,82,113]. Interestingly, the inhibition of cancer stem and initiating cells appears to be derivative of PP’s inhibition of the WNT [57] and mitochondrial pathways [73,82], both of which are specifically critical for cancer initiating and stem cells. Venugopal et al. showed that CD133^high^ and CD133-overexpressed cells displayed increased WNT signaling and were highly sensitive to PP treatment [57]. However, Xiang et al. [73] noted that although pyrvinium treatment showed selectivity to blast-phase chronic myeloid leukemia CD34^+^ progenitor cells, CK1α depletion did not affect PP sensitivity and WNT overexpression failed to rescue these cells. Instead, PP was shown to mainly accumulate in the mitochondria and the creation of chronic myeloid leukemia (CML) ρ0 cells that lacked mitochondrial DNA caused resistance to PP. Similar references to a mitochondrial-driven targeting of cancer stem cells were made by Datta et al. [82] in their assessment of PP in glioma-like stem cells. Recently, Dattilo et al. [114] showed that PP targeted triple-negative breast cancer stem cells through the inhibition of lipid anabolism, resulting in the increased death of cancer stem cells. Additionally, it seems as though the inhibition of stemness is not limited to cancer, but is also apparent in dysplastic tissues—as was described by Min et al. [71], showing that PP can inhibit Trop2^+^/CD133^+^/CD166^+^ dysplastic gastric mucosa stem cells in mouse and human organoids.

Interestingly, PP has been shown to promote wound repair, inhibit fibrotic tissue development [115,116,117], and protect from ischemic injury and promote healing [118,119]. This appears to be directly related to improved stem cell survival and the inhibition of inappropriate lineage commitment [120], which appears contradictory to the known efficacy of PP against cancer progenitor and stem cells [55,73,113]. However, it has been demonstrated that PP preferentially targets cancer progenitor cells compared to non-malignant normal progenitor cells [73]. This is likely due to the altered metabolism and reliance of cancer stem cells on mitochondrial-related pathways including oxidative phosphorylation [73,121] and lipid metabolism [114,122]. 

### 2.4. ELAVL1/HuR Inhibition

*ELAVL1*/HuR is an RNA-binding protein that has been shown to be an important post-transcriptional regulator of many cancer-associated pro-survival genes such as *VEGF*, *WEE1*, and *IDH1* [123,124,125,126,127]. Its mechanism, at least in part, is associated with the nucleocytoplasmic translocation and stabilization of 3′UTR of target mRNA transcripts [128,129]. Pyrvinium pamoate has been described by Goa et al. [84] as a functional HuR inhibitor that shifts the HuR cytoplasmic/nuclear equilibrium in favor of the nuclear import of HuR by blocking HuR nucleo-cytoplasmic translocation through inhibiting the checkpoint kinase1/cyclin-dependent kinase 1 pathway and activating the AMP-activated kinase/importin α1 cascade. 

It is important to note that a published byproduct of PP’s inhibition of mitochondrial function and ATP production is a relative increase in AMP and in AMP-related signaling, which raises the possibility that PP is not a direct activator of AMP-activated kinase/importin α1. Similarly, other studies have shown that the CRISPR knockout of HuR failed to rescue cancer cells from PP-induced cytotoxicity [78]. 

### 2.5. Androgen Receptor Inhibition

In 2008, PP was identified as a novel androgen inhibitor based on a high-throughput FRET screen [130]. It was shown to be a potent anti-prostate cancer agent that inhibited the function of androgen receptor (AR) splice variants, potentially through interaction with the AR DNA-binding domain (DBD), and the inhibition of several splicing factors (such as DDX17) [86,87,88]. Its DNA-binding domain interaction was thought to occur as a result of binding at the interface of the DBD dimer and the minor groove of the AR response element [88]. However, as noted by Li et al., PP treatment was toxic to LNCaP cells and AR-negative PC3 cells even at concentrations lower than the AR IC_50_, which is suggestive that PP’s anti prostate cancer efficacy was mediated through a non-AR-related mechanism [85].

### 2.6. Unfolded Protein Response 

Cells, and cancer cells in particular, undergo physiological processes that require a high rate of protein production and degradation. During protein production, unfolded or misfolded proteins can accumulate in the endoplasmic reticulum, resulting in increased cellular stress (i.e., ER stress). ER stress can also be increased upon physiological, environmental, or pharmacological insults. The unfolded protein response (UPR) is an adaptive cellular survival mechanism that regulates the cell’s handling of ER stress. In routine function, the ER stress signaling activates the unfolded protein response to reduce the volume of unfolded protein load and to help maintain normal cell function and metabolism [131]. However, chronic or extreme ER stress can trigger cell death by apoptosis and has been linked to many diseases such as cancer, diabetes, and neurodegenerative disorders [132,133,134,135,136]. 

Yu et al. [137] have shown that after hypoglycemia induced UPR, PP suppressed the transcriptional activation of UPR-regulated apoptosis and autophagy targets including XBP-1 and ATF-4 [83,137] along with inhibiting the induction of UPR transcription factors GRP78 (HSPA5) and GRP94 (HSP90B1). Interestingly PP did not inhibit the UPR when it was initiated by non-metabolic agents [137], potentially indicating that the inhibition of the UPR is secondary to the mitochondrial and metabolic dysregulation associated with PP treatment. Another aspect that is yet to be assessed is the impact of PP on the mitochondrial unfolded protein response (UPR^mt^), a crucial pathway for the maintenance of mitochondrial homeostasis [138].

### 2.7. Attenuation of Hedgehog Signaling

The hedgehog (SHH) signaling pathway is one of the key developmental pathways regulating cell proliferation, differentiation, and stem cell renewal [139]. Not surprisingly, the SHH pathway has been shown to play a key role in multiple cancer types [140]. Li et al. [141] were the first to describe PP as an inhibitor of the SHH pathway, and determined that PP treatment decreased the expression of downstream targets of the SHH pathway including Patched1 and Gli-family transcription factors in vitro and in vivo. Subsequently, El-Derhany et al. [142] noted that PP caused the dual inhibition of the WNT and SHH pathways. However, as a specific SHH target has yet to be identified, it is possible that due to the reciprocal coregulation between the WNT and SHH pathways that the inhibition of SHH is secondary to PP’s effects on the WNT pathway [143].

### 2.8. Inhibition of PD-1/PDL-1 Interaction 

Immunotherapies, and PD-1/PDL-1 inhibitors specifically, have revolutionized many fields of cancer therapy. PDL-1 (CD274)/PD1 (CD279) is an important modulatory/inhibitory immune checkpoint mechanism, utilized by some cancers to target and suppress the adaptive immune response [144]. Upon the binding of PDL-1, a transmembrane ligand, to its immune cell receptor PD-1 (which is highly expressed in activated T-cells), an inhibitory signaling cascade results in decreased T-cell proliferation, the induction of apoptosis in antigen-specific T-cells, and the inhibition of apoptosis in regulatory T-cells [144,145]. In a recent paper by Fattakhova et al. [146], a focused structure- and ligand-based drug library screen was employed that identified PP as an effective blocker of the PD-1/PDL-1 interaction. Of note, they found the IC_50_ for PP inhibition of PD-1/PDL-1 to be ∼30 μM, which is a promising starting point for PP to serve as a lead compound for the development of more potent inhibitors. However, since PP inhibits mitochondria, the WNT pathway and cancer cell viability at concentrations that are orders of magnitude lower than the noted IC_50_ for PD-1/PDL-1 inhibition, this immunoactive mechanism is likely not a relevant MOA for PP.

## 3. Delivery of Pyrvinium Pamoate

Current formulations of PP (Pyrvin, Vanquin, Pirok, Pyrcon, and Molevac) all deliver PP orally as either a suspension or in tablet formulation. Unfortunately, the systemic bioavailability of oral PP is uncertain. To date, there have been four published assessments of the systemic bioavailability of PP. In 1974, Buchanan et al. [147] assessed the systemic bioavailability of PP in humans after a single oral dose of PP at 5 mg/kg. They determined that 2/5 patients after 24 h had detectable levels of PP in the blood, while 2/4 patients had detectable levels of PP in the urine (concentrations not stated). In 1976, Smith et al. [16] followed up on this work and treated 12 patients with 350 mg of PP (this approximates the standard dose of 5 mg/kg with an average human male weight of ~70 kg), and assessed the amount of PP cleared through the urine or accumulating in the plasma. They determined that no patients displayed appreciable levels of PP accumulation in either plasma or urine. Both papers utilized PP fluorescence to assess the concentration of PP in their samples. While Buchanan et al. mentioned that they determined that the limits of detection of the assay were 5 and 11 ng in urine and plasma, respectively (presumably per 10 mL of sample as stated), in Smith’s work, where concentrations were stated, they demonstrated pre-drug concentrations of PP of 20.19 µg in urine (sample of 500 mL or less, ~70 nM or more) and 3.9 ng/mL (~6.77 nM) in plasma, significantly higher than the limits of detection as described in the work by Buchanan et al. Therefore, the results from both studies describing the bioavailability of PP in plasma or urine in humans are questionable, especially given the lack of positive controls or animal work with PP dosed intraperitoneally (IP) or intravenously (IV) to ensure uptake. This is particularly relevant as PP has frequently been demonstrated to have anti-cancer IC_50_s between 10 and 100 nM [78]. 

In more recent work utilizing mass spectrometry to quantify PP, it has been shown that PP can be detected in the plasma and organs even when dosed orally. In 2009, Jones et al. [86] detected up to 40.2 ng/mL (~69.8 nM) in mouse plasma after the oral dosing of PP at 5 mg/kg. Subsequent work by Schultz et al. in 2021 [78] assessed PP levels in fat, muscle, pancreatic tissue and plasma at multiple timepoints between 15 min and 6 h after dosing either IP, IV, or orally at 5, 20, or 35 mg/kg. Unfortunately, while this work was more expansive, it was less conclusive with high variability. Significant accumulations of PP were detected in fat (highest concentration 57 ng/mL or ~100 nM) and pancreas tissues (highest concentration 52 ng/mL or ~90 nM) after the oral dosing of PP during timepoints where PP was undetectable in the plasma. This indicated that PP may preferentially accumulate in fat and fatty tissues and may have a high volume of distribution. Taken together, these data indicate that there is some systemic bioavailability of PP when dosed orally in mice; this is supported by numerous papers that have demonstrated the systemic efficacy of PP against various cancer models in mice when dosed orally [45,47,78,86,137,148]. There is currently an ongoing trial attempting to determine the safety and bioavailability of higher doses of oral PP (NCT05055323 [46]).

Beyond the potential to deliver PP as currently formulated as an oral therapeutic, there have been several attempts to utilize PP or pyrvinium in other formulations or with alternative delivery methods. An entirely novel method for the delivery of PP focused on utilizing cyclodexterin-based polymer systems in order to release PP in a controlled long-term fashion (up to 30 days of continuous release) [149]. Rohner et al. describe how this could be useful for utilizing PP to aid in wound healing and to inhibit fibrotic tissue development. This methodology could also be potentially utilized to allow PP to target topical fungal infections or even potentially skin cancer nodules. A novel systemic methodology for delivering PP by Hatamipour et al. relied upon the liposome-based delivery of PP [150]. This methodology demonstrated some potential and did have efficacy in a melanoma mouse model (improved median survival, 14 days control vs. 20 days free PP vs. 23 days liposomal PP). However, due to the therapy being delivered IV, it is unclear whether this methodology represents an improvement upon PP being dosed orally. The effects could potentially be improved if another cell line was assessed in vivo that was more sensitive to PP, as the B16-F0 cells utilized in this work had a high IC_50_ in vitro (8.3 µM, while many cancer cell lines have IC_50_s in the low nanomolar range) or in combination with another agent such as abemaciclib or salirasib [49,151]. 

While pyrvinium pamoate has been in use for over 60 years, between 1947 and 1959, pyrvinium was primarily utilized in humans as pyrvinium chloride. Pyrvinium chloride has an improved water solubility profile compared to pyrvinium pamoate, and thus likely increased systemic distribution. Furthermore, pyrvinium chloride has been utilized in hundreds of patients with acceptable toxicity profiles [11,12,13,15]. Unfortunately, due to the vast body of existing literature, it may be quite difficult to commercialize pyrvinium chloride for the treatment of cancer. While low patentability potential and limited profitability do not affect a drug’s potential clinical implications, they do drastically affect the ability to acquire funding for testing drugs in the clinic. Therefore, even simply repurposing potentially useful existing therapeutics can be exceedingly expensive and difficult. 

Pyrvinium tosylate has also been utilized as an anticancer agent [61,69]. Pyrvinium tosylate presumably has greater water solubility due to the tosylate counterion similar to pyrvinium chloride, and thus is likely to have increased bioavailability as well [152]. However, pyrvinium tosylate has not been extensively utilized in humans and faces similar challenges of patentability to both pyrvinium pamoate and pyrvinium chloride.

## 4. Perspectives and Future of PP as a Therapeutic

There is clear and evident utility for the use of PP for the treatment of many human diseases including viral, bacterial, protozoal, fungal, and even multicellular parasitic infections. Perhaps most strikingly, PP is a potent anticancer therapeutic with exquisite efficacy against multiple cancers. PP may even be a potential prophylactic for high-risk individuals such as those with familial adenomatous polyposis [153]. With such clear evidence for the efficacy of PP, the logistic and economic obstacles to using this agent in the clinic are unfortunate. Both PP and pyrvinium chloride have been used in hundreds of thousands of human patients and have been found to be incredibly safe agents. PP has been shown to be effective in multiple cancer types in vivo in many publications (Table 1), demonstrating clear potential for efficacy in humans. There is a lack of clinical interest in PP, which appears to be due to poor patentability prospects. Phase I, II, and III clinical trials cost ~4, 13, and 20 million dollars, respectively [154]. Due to the wealth of previously published literature regarding PP as an anticancer agent as detailed in this review, creating an enforceable patent strategy for PP would be challenging. Without a patent, it would be difficult for any organization running a clinical trial to recoup their investments. To date, there is only a single clinical trial assessing PP in cancer patients (NCT05055323 [46]). Importantly, this is an investigator-sponsored trial that is being supported by Thomas Jefferson University in an attempt to demonstrate that PP is safe for the treatment of cancer, with the secondary objectives of determining pharmacokinetic and pharmacodynamic profiles.

Therefore, the likely limitations and strengths of pyrvinium pamoate, pyrvinium chloride, or novel pyrvinium pamoate formulations are as follows: (1) pyrvinium pamoate has unknown and potentially low oral systemic distribution, and low patentability; however, PP has the lowest barrier for use in humans due to extensive historic safety data. (2) Pyrvinium chloride likely has higher oral systemic distribution, although unproven. Similar to PP, it has low patentability, and while it has been utilized in humans, pyrvinium chloride has not been well studied for cancer. Although pyrvinium chloride’s efficacy is presumably comparable to PP, it is not currently manufactured as a Current Good Manufacturing Practices (CGMP) human pharmaceutical. (3) Novel PP formulations or derivatives have the highest potential for patentability and thus securing funding to drive clinical trials. However, they would have the longest time to the clinic and highest cost involved in reaching the clinic, and in particular would face significantly greater barriers in demonstrating safety compared to PP and pyrvinium chloride. 

While each method for bringing PP to the clinic presents its own unique challenges, it is clear from the wealth of previous literature that any or all of these routes could be highly impactful for patient care if successfully employed.

## Figures and Tables

**Figure 1 biomedicines-10-03249-f001:**
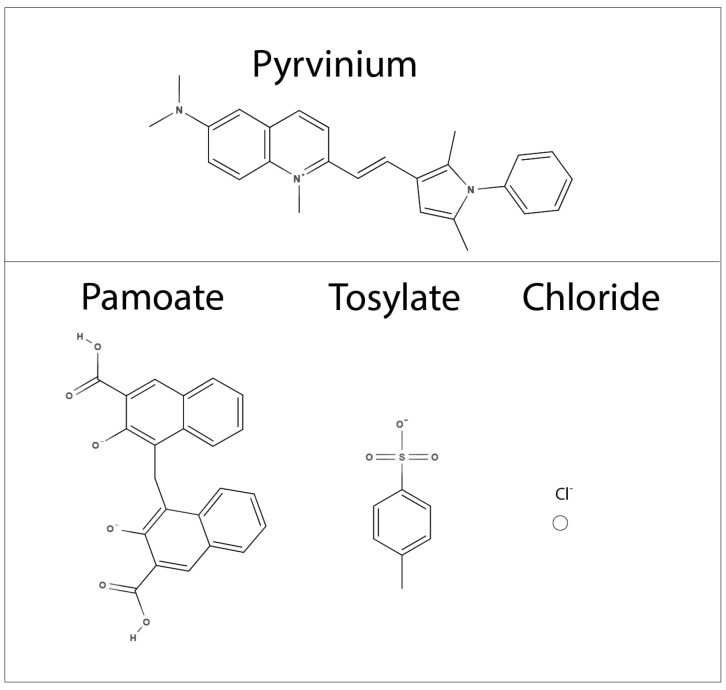
Chemical structure of pyrvinium with its salts counterions: pamoate (embonate), tosylate, and chloride.

**Figure 2 biomedicines-10-03249-f002:**
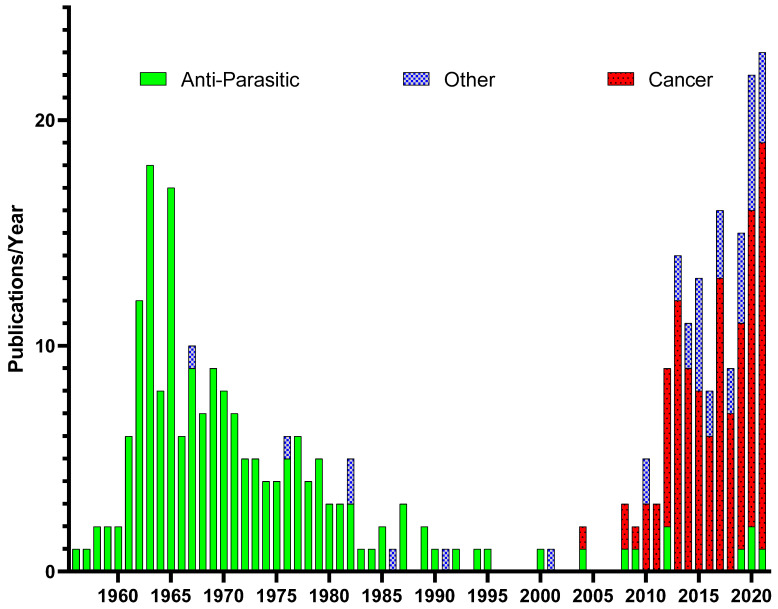
Shifts in pyrvinium pamoate publications over time: PP has been found to be effective in multiple models of human diseases. PP publications/year were broken down by disease type, finding a focus mainly in the anthelmintic activity of PP prior to 2004 and a shift of thematic interest in PP as an anti-cancer agent after 2004.

**Table 1 biomedicines-10-03249-t001:** Pyrvinium pamoate as an anti-cancer agent: Instances of PP as an anticancer agent were assessed by cancer type, whether it was used in vitro or in vivo, and what mechanism of action the authors claimed PP had in their cancer type/model. * is indicative of publications using pyrvinium tosylate as opposed to pyrvinium pamoate.

Mechanism of Action	Cancer Type	In Vitro	In Vivo
WNT	Intestinal adenomas		[47]
WNT	Colorectal	[48,49,50,51,52]	[50]
WNT	Synovial sarcoma	[53]	
WNT	Breast	[54]	[54]
WNT	Lung	[55,56]	
WNT	Glioblastoma	[57,58]	[57,58]
WNT	Ovarian	[59,60]	[59,60]
WNT	MDS	[61]	[61]
WNT	Cervical	[62]	[62]
WNT	Wilms tumor	[63]	[63]
WNT	Uveal melanoma	[64]	
WNT	Clear cell RCC	[65]	[65]
WNT	Multiple myeloma	[66,67]	
WNT	Malignant mesothelioma	[68]	
WNT	Nasopharyngeal carcinoma	[69] *	
WNT	Myelodysplastic syndrome		[61] *
WNT	Osteosarcoma	[70]	
WNT	Gastric adeno	[71]	
WNT	Glioblastoma	[58]	
AKT/mTOR	Pancreatic	[45]	[45]
AKT/mTOR	Colorectal	[45]	[72]
Mitochondria	CML	[73,74]	[73]
Mitochondria	Multiple myeloma	[66,75]	
Mitochondria	AML	[76]	
Mitochondria	Pancreatic	[77,78]	[78]
Mitochondria	Hepatic	[77]	
Mitochondria	Colorectal	[77]	
Mitochondria	Lymphoma	[79]	
Mitochondria	B cell ALL	[80]	
Mitochondria	Melanoma	[81]	
Mitochondria	Glioma	[82]	
Mitochondria	AML	[83]	
HuR (through AMPK)	Pancreatic	[78]	[78]
HuR (through AMPK)	Bladder	[84]	[84]
Unfolded protein response	AML	[83]	
DNA-binding domain of the human androgen receptor	Prostate	[85]	
DNA-binding domain of the human androgen receptor	Prostate	[86,87,88]	[86,88]

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
