# Peer review of "Pyrvinium Pamoate: Past, Present, and Future as an Anti-Cancer Drug"

_biomedicines, 2022, doi:10.3390/biomedicines10123249_

Round 1

Reviewer 1 Report

This review article focuses on Pyrvinium pamoate (PP) a very old molecule used at first essentially as an anthelmintic. Over the past two decades evidence has emerged showing this molecule to have strong anti cancer activity in various human cancers. Authors go through the story of the molecule  analyzing what is known about PP multiple anti-cancer mechanisms of action and the challenges associated with repurposing pyrvinium as anti-cancer agent

Overall, this review is well written and exhaustive. Only minor weaknesses should be addressed. 

1-The inclusion of the Pyrvinium pamoate chemical structure as well as of a sketch to summarize PP most important cellular mechanisms of action as an anticancer agent would make this review more appealing.

2-page 5/130 “again” should be “against”

3-page 6/171 please explain the acronym “CML”

4- page 6/190: please introduce briefly HuR 

5- page7/244 please introduce briefly PD-1/PDL-1 

6-page 7/252 “amechanism” should be “mechanism”

7-page 8/273 please explain IP and IV acronyms

Author Response

We thank the reviewer for the insightful comments, which improve the clarity and readability of our manuscript.

Accordingly, we have added a figure detailing the chemical structure of pyrvinium and its salts.

We have corrected the misspelled words and elaborated some more on HuR and PD1/PDL1.

Reviewer 2 Report

The manuscript presented for review is a relevant, very interesting, and well-designed work.

I believe that this work is worthy to be published in the journal Biomedicines in the submitted form and the manuscript does not require any additional changes. However, there are some technical typos in the manuscript that should be corrected, and that is the only reason why my decision is "accepted after minor revisions", but such a decision absolutely does not reduce the high level and value of this work!

Points that authors should correct:

1. Name: write all words in capital letters and remove the dot at the end

2. Line 117 - put a space before the reference (10)

3. Correct the spelling of "in vivo" in lines 103, 305, and 333

4. Line 356 - put a comma after "particular"

Author Response

We thank the reviewer for the insightful comments, which improve the clarity and readability of our manuscript.

Accordingly, we have revised the title, kept 'in-vivo' and 'in-vitro' consistently hyphenated and italicized, and added the spacing and comma as requested.